# Role of the ROS-JNK Signaling Pathway in Hypoxia-Induced Atrial Fibrotic Responses in HL-1 Cardiomyocytes

**DOI:** 10.3390/ijms22063249

**Published:** 2021-03-23

**Authors:** Chin-Feng Tsai, Shun-Fa Yang, Chien-Hsien Lo, Hsiao-Ju Chu, Kwo-Chang Ueng

**Affiliations:** 1Division of Cardiology, Department of Internal Medicine, Chung Shan Medical University Hospital, School of Medicine, Chung Shan Medical University, Taichung 40201, Taiwan; csy230@csmu.edu.tw (C.-F.T.); cshy1420@csh.org.tw (C.-H.L.); 2Institute of Medicine, Chung Shan Medical University, Taichung 40201, Taiwan; ysf@csmu.edu.tw; 3Department of Medical Research, Chung Shan Medical University Hospital, Taichung 40201, Taiwan; pig191919@seed.net.tw

**Keywords:** hypoxia, fibrosis, atrial fibrillation, signal transduction

## Abstract

By promoting atrial structural remodeling, atrial hypoxia contributes to the development of the atrial fibrillation substrate. Our study aimed to investigate the modulatory effect of hypoxia on profibrotic activity in cultured HL-1 cardiomyocytes and explore the possible signaling transduction mechanisms of profibrotic activity in vitro. Hypoxia (1% O_2_) significantly and time-dependently increased the expression of hypoxia-inducible factor (HIF)-1α and fibrotic marker proteins collagen I and III (COL1A and COL3A), transforming growth factor (TGF)-β1 and α-smooth muscle actin (SMA). Western blot or immunohistochemistry analysis showed that hypoxia-induced increase in COL1A and COL3A was significantly attenuated by the addition of SP600125 (a specific c-Jun N-terminal kinase [JNK] inhibitor) or expression of dominant-negative JNK before hypoxia treatment. The inhibition of hypoxia-activated phosphorylation of JNK signal components (JNK, MKK4, nuclear c-Jun and ATF-2) by pre-treatment with SP600125 could suppress hypoxia-stimulated HIF-1α upregulation and fibrotic marker proteins expression. Hypoxia significantly increased reactive oxygen species (ROS) production in cultured HL-1 atrial cells. Pre-treatment with N-acetylcysteine significantly abrogated the expression of nuclear HIF-1α, JNK transduction components and fibrotic marker proteins. Taken together, these findings indicated that the hypoxia-induced atrial profibrotic response occurs mainly via the ROS/JNK pathway, its downstream upregulation of HIF-1α and c-Jun/ATF2 phosphorylation and nuclear translocation to up-regulate the expression of fibrosis-related proteins (COL1A, COL3A, TGF-β1 and α-SMA). Our result suggests that suppression of ROS/JNK signaling pathway is a critical mechanism for developing a novel therapeutic strategy against atrial fibrillation.

## 1. Introduction

Atrial fibrillation (AF) is the most common sustained arrhythmia clinically. Extensive evidence indicates that atrial fibrosis is an important contributor to the AF substrate. Atrial fibrosis is characterized by abundant accumulation of matrix proteins in extracellular space, increased expression of profibrotic cytokines (e.g., transforming growth factor [TGF]-β1 and α-smooth muscle actin [SMA]) and more pronounced simultaneous expression of collagen I and III (COL1A and COL3A) [1]. Cardiomyocytes have recently been implicated as an important source of extracellular matrix and cardiac fibrotic regulators such as TGF-β1 expressed in response to a specific environmental stress [2]. However, the regulatory mechanisms and signaling pathways involved in the development of atrial fibrosis are still not completely understood.

Atrial ischemia is an important determinant of AF substrate development in myocardial infarction. In a canine model of chronic coronary artery occlusion, Nishida et al. found that sources (conduction abnormalities) of sustained re-entry at the border of areas of atrial infarction were associated with significant peri-infarct fibrosis [3]. Tissue hypoxia is an essential feature of ischemia. Hypoxia is a stimulus to collagen synthesis in cardiac fibroblasts, thereby possibly contributing to the development of myocardial fibrosis and enhancing the propensity to arrhythmia occurrence [4]. Hypoxia or ischemia can upregulate hypoxia-inducible factor-1α (HIF-1α), a transcriptional factor playing a central role in the maintenance of oxygen hemostasis. Recent reports have suggested that HIF-1 plays a role in the cardiorespiratory response to hypoxia [5]. Atrial hypoxia and the HIF pathway have been implicated in structural changes of the atria, such as cardiac fibrosis, that may lead to the development of AF [6,7]. It has been reported that HIF-1α gene expression increases transiently in parallel with fibrogenesis in atrial myocardium during the early response of cardiomyocytes to AF [6]. In response to hypoxia, the heart also generates reactive oxygen species (ROS), resulting in cardiomyocyte adaptation through various signaling cascades [8]. Accumulating evidence has implicated a potential role of oxidative stress in the pathogenesis of AF [9]. Excessive production of ROS is likely involved in the structural and electrical remodeling of the heart, contributing to fibrosis and thrombosis [10]. However, the involved underlying molecular mechanisms and signal pathways remain unclear. Recent evidence suggests that atrial fibrosis is promoted by the activated renin-angiotensin-aldosterone system and enhanced extracellular signal-regulated protein kinase (ERK) pathway in human and animal models of AF [11,12]. Progressive interstitial changes provoked by mitogen-activated protein kinase (MAPK), coupled to the angiotensin II regulatory pathway, increase the risk of AF [11]. We have reported that elevated mineralocorticoid receptor activity plays a central role in aldosterone-mediated activation of the MAPK signaling pathway and subsequent profibrotic effects in HL-1 atrial cells [12]. Thus, the present study investigated their relevance and the possible molecular mechanism underlying the upregulation of HIF-1α following hypoxia induction of atrial fibrosis. The purpose of our study was two-fold: 1) to examine the effect of hypoxia stimulation on the expression of HIF-1α, MAPKs signaling molecules and fibrosis-related proteins in cultured HL-1 atrial myocytes; 2) to determine the involvement of the ROS and MAPK signaling pathways in the mechanism underlying the upregulation of HIF-1α and hypoxia-induced atrial fibrosis.

## 2. Results

### 2.1. Effect of Hypoxia on the Expressions of HIF-1α and Profibrotic Proteins

To evaluate the effect of hypoxia on atrial fibrosis-related proteins, we examined the time-dependency of the effect of hypoxia (1% O_2_) on the expression of HIF-1α and profibrotic proteins (COL1A, COL3A, TGF-β1 and α-SMA) in HL-1 cells over time (3, 6, 9 and 24 h of hypoxia). HIF-1α expression was significantly increased after 3 h of hypoxia exposure and continued to increase at a steady rate for 24 h, reaching on average 1.4-fold above baseline (*p* < 0.05). Western blotting showed that the expressions of COL1A and TGF-β1 protein were significantly increased at 6 h and of COL3A and α-SMA proteins significantly induced at 3 h after hypoxia treatment. The peak at which expression of COL1A, COL3A (the major cardiac fibrotic markers), TGF-β1 (a critical profibrotic factor) and α-SMA (a trans-differentiation marker) were significantly increased were 1.93-fold, 1.31-fold, 1.40-fold and 1.65-fold (*p* < 0.01), respectively (Figure 1A). These results support the hypothesis that hypoxia treatment can up-regulate expression of profibrotic proteins in cultured HL-1 atrial myocytes.

To further demonstrate the effects of hypoxia on the expression of HIF-1α and profibrotic proteins, HL-1 atrial cells were treated with the well-known hypoxia-mimetic agent, CoCl_2_ (a prolyl hydroxylase inhibitor) for 6 h. Compared with 1% O_2_, CoCl_2_ significantly, although to a lesser extent, increased the expression of HIF-1α and profibrotic proteins (TGF-β, COL1A, COL3A and α-SMA) in HL-1 atrial cells (Figure 1B).

### 2.2. Hypoxia-Induced Atrial Fibrotic Response via JNK-Dependent Pathways

To identify the possible signaling pathways mediating expression of the hypoxia-induced fibrosis gene, HL-1 cells were treated with different MAPK pathway inhibitors, SB203580 (a p38 inhibitor), U0126 (a specific MAPKK/MEK inhibitor of ERK activation) and SP600125 (an inhibitor of the JNK phosphorylation pathway) 30 min before hypoxia treatment. Pretreatment with SB203580 significantly inhibited only hypoxia (1% O_2_ for 6 h)-induced expression of COL3A and TGF-β1 and pretreatment with U0126 significantly inhibited only hypoxia-induced expression of HIF-1α and COL1A. However, pre-treatment with JNK inhibitor (SP600125) significantly and consistently abrogated the expression of HIF-1α and all profibrotic proteins (COL1A, COL3A, αSMA and TGF-β1), suggesting that JNK pathway activation is involved in the hypoxia-induced upregulation of the profibrotic proteins expression in HL-1 cells (Figure 2A).

In confirmation of the signaling pathway responsible for hypoxia-induced atrial profibrotic proteins expression, we found that hypoxia treatment, with low oxygen or CoCl_2_ for 6 h, can up-regulate the JNK signaling pathway. Hypoxia stimulated expressions of phosphorylated JNK and SEK/MKK4 (a protein kinase that activates JNK in response to environmental stress), suggesting that the JNK pathway probably acts as an upstream mediator of hypoxia-induced atrial fibrosis signaling (Figure 2B). We examined the time-dependency of the effect of hypoxia (1% O_2_) on the expression of the JNK signaling pathway in HL-1 cells over time (3, 6, 9 and 24 h of hypoxia). Phosphorylated JNK and SEK/MKK4 expression were significantly increased after 3 h of hypoxia exposure and continued to increase at a steady rate for 24 h (Figure 2C). We further verified the role of the JNK pathway in atrial pro-fibrogenesis, by showing that hypoxia for 6 h in HL-1 atrial cells significantly increased the expression of HIF-1α, all profibrotic proteins (COL1A, COL3A, TGF-β1 and α-SMA) and phosphorylated JNK and MKK4 proteins (Figure 3A). Treatment with SP600125 30 min before hypoxia significantly attenuated this increase, indicating that an upstream activator of the JNK signaling pathway, SEK/MKK4, was involved in modulation of profibrotic protein expression (Figure 3A).

To further confirm that JNK activity was involved in hypoxia-induced profibrotic proteins expression in HL-1 cells, we investigated whether expression of dominant-negative JNK could affect hypoxia-induced profibrotic proteins expression. Transfection of dominant-negative JNK had no effect on the expressions of COL1A and COL3A proteins by HL-1 cells when compared to non-transfected cells but did significantly attenuated the hypoxia-induced expression of COL1A and COL3A proteins and phosphorylated JNK when the Hl-1 cells were transfected 24 h before hypoxia treatment (1% O_2_ for 6 h) (Figure 3B). Furthermore, the JNK pathway modulation of the hypoxia-induced expression of fibrotic marker was validated by using immunofluorescence analysis to compare the number of pixels labeled with fibrosis marker protein (COL1A and COL3A) signals in hypoxia-exposed HL-1 cells either pre-treated or not with JNK inhibitor SP600125. The number of pixels (% of control) was significantly lower in hypoxia-induced HL-1 cells pre-treated with SP600125 (Figure 4A,B). Since the JNK inhibitor could modulate COL1A and COL3A expression, we further investigated the involvement of downstream signal transduction of the JNK pathway on the hypoxia-treated HL-1 cells.

It has been reported that HIF-1α nuclear translocation acts as a master regulator of numerous hypoxia-inducible genes under hypoxic conditions [8]. The transcription factors, c-Jun and activating transcription factor (ATF)-2α, have been identified as substrates for JNK activity. Stress-activated protein kinase/c-Jun NH(2)-terminal kinase (SAPK/JNK), when active as a dimer, translocate to the nucleus and regulate transcription through their effects on c-Jun, ATF-2 and other transcription factors [13]. In our study, HIF-1α translocated and accumulated in the nuclei of cells under low oxygen conditions. As shown by Western blot analysis, hypoxia (1% O_2_) and CoCl_2_ treatment for 6 h significantly upregulated the levels of nuclear HIF-1α, phosphorylated c-Jun and ATF-2 protein in HL-1 atrial cells (Figure 5A). As shown in Figure 5B, SP600125 treatment did not affect the nuclear HIF-1α expression but significantly decreased the expression of nuclear phosphorylated c-Jun and ATF2 compared to the control group in HL-1 cells. The addition of SP600125 at 30 min before hypoxia treatment significantly attenuated nuclear HIF-1α expression and eliminated the expression of nuclear phosphorylated c-Jun and ATF2 induced by hypoxia (1% O_2_ for 6 h) (Figure 5B). These results suggested that low oxygen stress could activate nuclear HIF-1α and induce SAPK/JNK downstream signaling phosphorylation of c-Jun and ATF2, resulting in efficient activation of the target genes controlled by the JNK-regulated transcription factors.

### 2.3. ROS Modulate Hypoxia-Induced Atrial Profibrotic Proteins Expression

Reactive oxygen species (ROS) have been implicated in the pathogenesis of a variety of diseases such as atrial fibrillation and play a role as second messengers that regulate mitogenic signal transduction in rat cardiac myocytes [14,15]. Using dichlorofluorescein diacetate (DCFH-DA) to determine the hypoxia-induced ROS changes in the cellular redox environment, we found a significant difference in ROS production between under hypoxic and normoxic conditions. The level of ROS was greater in hypoxic cells. Cells pre-treated with N-acetyl cysteine (NAC) (an ROS scavenger) significantly inhibited ROS production induced by hypoxia (Figure 6A). Furthermore, we sought to elucidate the effect of ROS on the change in hypoxia-induced atrial profibrotic proteins and on the phosphorylation of selected signaling components of the JNK pathway. As shown by Western blot analysis in Figure 6B–D, NAC treatment for 2 h had no effect on the normal expression of nuclear HIF-1α, profibrotic proteins (COL1A, COL3A, TGF-β1, α-SMA) and the phosphorylation of JNK signaling components (JNK, c-Jun, ATF-2) in HL-1 cells, but did significantly abrogate the hypoxia-induced expression of nuclear HIF-1α, profibrotic proteins (COL1A, COL3A, TGF-β1, α-SMA) and the phosphorylated JNK signal components (JNK, MKK4, c-Jun, ATF-2) (Figure 6B–D). However, NAC treatment significantly increased the protein expression of pMKK4 compared to the control group. These results suggest that ROS are involved in the hypoxia-induced atrial profibrotic remodeling. The increased ROS levels that activate HIF-1α activation and JNK signaling pathways also increase the profibrotic proteins expression induced by hypoxia in cultured HL-1 atrial cells. Altogether, our results strongly suggested that hypoxia promotes atrial fibrosis mainly via the ROS/JNK pathway and downstream activation of nuclear c-Jun/ATF2 phosphorylation to up-regulate the expression of fibrosis-related proteins (COL1A, COL3A, TGF-β1, α-SMA) (Figure 7).

## 3. Discussion

Hypoxia is known to increase AF risk. Gramley et al. reported that AF is closely associated with an atrial upregulation of hypoxic and angiogenic markers in human atrial tissues [16]. Experimental atrial ischemia creates a substrate for AF maintenance, apparently by causing local conduction slowing that promotes reentry. In this model, atrial fibrosis is likely an important factor in stabilizing reentry and promoting AF [3]. Thus, our study was designed to investigate the potential profibrotic impact of hypoxia on cultured HL-1 atrial myocytes. The cellular response to hypoxia is primarily mediated by the heterodimeric transcription factor HIF-1. HIF-1α has been closely associated with renal, liver and pulmonary fibrosis disease [17,18,19]. Previous studies also showed that the increase in HIF-1α gene expression in the myocardium may be involved in structural changes in the myocardium, including atrial fibrosis [6]. In isoproterenol-treated rats, Su et al. showed positive correlation of HIF-1α expression with the extent of myocardial fibrosis (collagen volume fraction index), which implies that HIF-1α can facilitate the expression of matrix metalloproteinase-9 (MMP-9) and TGF-β1 and thus, induce atrial fibrosis [7]. In our study, hypoxia significantly upregulated HIF-1α and profibrotic proteins expression in a time-dependent manner. Under hypoxic conditions, the peak of HIF-1α expression at 3 h preceded the peak of profibrotic proteins expression at 6 h, suggesting that these fibrotic marker proteins may be associated with the HIF-1α activation following hypoxia. Given that cells and organs need to adapt to changes in oxygen supply, it would not be surprising that a significant number of HIF-1α target genes are regulated in a tissue-specific manner [17,18,19,20]. HIF-1α is found to mediate TGF-β1-induced expression of fibrogenic proteins such as collagen and plasminogen activator inhibitor in renal epithelial cells and in alveolar macrophages [17,19]. In a bile duct ligation-induced liver fibrosis animal model, Moon et al. demonstrated that collagen I and α-SMA expressions in fibrotic livers were significantly decreased after HIF-1α was deleted [20]. Blocking HIF-1α transcription with a biochemical inhibitor or gene-specific knockdown decreased basal and TGF-β1-stimulated collagen I expression. Therefore, it pointed out that targeting HIF-1α is a therapeutic strategy against fibrosis [21]. Our data allow us to postulate that hypoxia via the HIF pathway may play a role in the development of atrial fibrosis; however, we could not determine whether the upregulation of HIF-1α is required for the induction of profibrotic proteins during hypoxia without silencing HIF-1α prior to hypoxia.

The MAPKs are widely involved in vital cellular processes and, also play critical roles in responding to cellular stress. p38 MAPK and JNK, known as stress-activated protein kinase, are primarily activated by noxious environmental stimuli, e.g., hypoxia, whereas ERK1/2 are primarily stimulated by mitogenic and differentiative factors [13,22]. Previous experimental and clinical studies have shown that MAPKs pathways play an important role in cardiac fibrosis [11,12,23,24,25]. Here, we demonstrated that hypoxia treatment can remarkably induce SEK/MKK4 expression and phosphorylation of the stress kinase JNK, which positively modulated HIF-1 α and profibrotic proteins expression in cultured HL-1 cells. The inhibition of hypoxia-induced phosphorylation of JNK by pretreatment with the JNK pathway inhibitor (SP600125) could suppress hypoxia-stimulated HIF-1α activation and profibrotic proteins expression examined by Western blot or immunofluorescence, which means that hypoxia-induced expression of profibrotic proteins in cultured HL-1 cells linked to the activation of the JNK pathway. Transfection of the dominant-negative JNK construct also inhibited hypoxia-induced expression of the fibrotic markers COL1A and COL3A. However, coincubation of p38 kinase inhibitor (SB203580) or ERK inhibitor (U0126) had no consistent effects on profibrotic proteins expression. In the heart, enhanced p38 MAPK signaling has been implicated in cardiac hypoxic and ischemic injury [26]. However, in our study, the JNK pathway was the main pathway involved in hypoxia-induced expression of profibrotic proteins in HL-1 cells. Different MAPK pathways may be involved in different cell types (atrial versus ventricular or vascular smooth muscle cells) or different stimuli in the regulation of profibrotic proteins expression. Our data are inconsistent with previous studies that implicate JNK as a signal for the cardiac remodeling process. In one study, deletion of JNK1 in the heart resulted in an increase in fibrosis following pressure overload [27]. Similarly, chronic treatment with a JNK inhibitor led to increased cardiac fibrosis and apoptosis in the cardiomyopathic hamster model [28]. On the other hand, in a model of angiotensin-II-induced hypertrophy, mice deficient for ASK1-JNK signaling demonstrated an attenuation of cardiac fibrosis and remodeling [29]. In vitro, one study showed that JNK activation in H9c2 cardiomyoblasts resulted in the upregulation of MMP-2 (but not MMP-9) activity [23]. Likewise, in vivo studies on the loss of β1-integrins showed increased JNK activity was associated with increased MMP-2 activity, which corresponded with cardiac fibrosis remodeling in this setting [30]. Collectively, these results may indicate that JNK’s role in cardiac pathological remodeling may depend on the activating stimuli. Here, we showed that hypoxia (1% O_2_) treatment increased the phosphorylation of JNK and accentuated markers associated with fibrosis. Furthermore, JNK inhibition could suppress hypoxia-stimulated profibrotic proteins expression, indicating that the hypoxia-induced JNK pathway activation upregulates profibrotic proteins expression in HL-1 atrial cells. Moreover, hypoxia upregulated not only JNK but also phosphorylation of its downstream effectors, c-Jun and ATF-2α kinase, with concordant increase of HIF-1α in the nucleus. ATF-2 is a cAMP response element binding protein and normally regulates and heterodimerizes with other transcription factors, including c-Jun and c-fos [31]. JNK can promote transcription of genes involved in renal fibrotic responses through phosphorylation of c-Jun and ATF2 [32]. Studies have shown that JNK dependent phosphorylation of ATF2/c-Jun transcription factors resulting in TGF-β transcription play the important roles in the formation of oral submucous and renal fibrosis [33,34]. Because these proteins are shown to colocalize in distinct compartments of the nucleus, we suggest that these stress-responsive transcription factors communicate in close proximity to each other, thereby enabling the fine tuning of the cellular response for adaptation. Hence, our study conclusively showed that the JNK pathway is important in regulating HIF-1α and profibrotic proteins expression in cultured HL-1 cells exposed to hypoxia stress.

Oxidative stress under hypoxia and the subsequent generation of ROS are thought to represent an important mechanism underlying adverse cardiac remodeling. Indeed, oxidative stress is increased in heart failure, hypertension, cardiac fibrosis and hypertrophy [9,10,35]. In angiotensin-II treated rats, Zhao et al. showed that increased deposition of collagen was dependent on cardiac oxidative stress (upregulated NADPH oxidase) and antioxidants given simultaneously with angiotensin II reduced the myocardial collagen volume and prevented the expression of TGF-β1 [10]. Moreover, the involvement of oxidative stress in the activation of pro-fibrotic cascades in cardiac cells has been observed in mice with cardiac deficiency or overexpression of the NADPH-oxidase subunit Rac-1 [36,37]. Similarly, ROS are also involved in hypoxia-induced cardiac fibrosis. The harmful oxidant effects of ROS are linked to several key redox-sensitive signaling pathways in fibrogenesis [38]. Additionally, hypoxia-induced ROS generation has been shown to be both necessary and sufficient to stabilize and activate HIF-1α during hypoxia [39]. In this context, our study showed that hypoxia-induced ROS production acted as an upstream regulator of JNK/ATF2, which positively correlated the expression of HIF-1α and profibrotic proteins in hypoxia-treated HL-1 cells. Hypoxia increased ROS formation in cultured HL-1 cells. Pre-treatment of HL-1 cells with NAC (ROS scavenger) abrogated the activity of the hypoxia-induced JNK pathway including the upstream activator (MKK4) and downstream effector (ATF2, c-Jun), expression of nuclear HIF-1α and upregulation of profibrotic proteins expression. This study indicates that ROS-mediated JNK activation by hypoxia (1% O_2_) triggers intracellular signals that lead to the upregulation of expression of HIF-1α and profibrotic proteins. Previous studies showed that hypoxia induces mitochondrial ROS levels that may serve as second messengers in the triggering of signaling cascades [15]. ROS-dependent accentuation of JNK MAPK signaling is well established in cancer research. Here, we conclusively reported that the activation of the JNK pathway is a downstream consequence of ROS induction during hypoxia. This finding is inconsistent with previous studies that showed redox-dependent MAPK signaling pathways play a major role in the development of cardiac fibrosis [36,40]. Shih et al. recently demonstrated that TGF β1-induced cardiac fibroblast activation and proliferation promote cardiac fibrosis by enhancing redox-sensitive JNK signaling [38]. Moreover, recent studies show that ROS regulates activation of the nuclear transcriptional factor ATF2 by phosphorylated JNK [41] and increases ROS levels activate upstream HIF-1α pathways, such as MAPK signaling, which can in turn induce HIF-1α dependent transcriptional activity in cancer cells and HL1 cardiomyocytes [42]. These findings are in agreement with the results of the present study. Because ROS scavenging and JNK pathway inhibition abrogated hypoxia-mediated profibrotic proteins upregulation in the present study, we conclude that the upregulation of profibrotic proteins following hypoxia occurred through an ROS- and JNK-signaling pathway activation.

### Study Limitations

In cardiovascular disease, collagen mRNA or protein expression is often different from collagen volume fraction because collagen synthesis depends on the time course of the disease and the underlying disease itself. Therefore, whether results in cultured HL-1 cells can be readily extrapolated to atrial tissues impaired by different cardiovascular diseases is unknown. It is also unclear whether hypoxia-mediated signaling changes with the extent of atrial remodeling. The reduction of JNK and ROS has been verified to improve fibrosis in cardiomyocytes. Increasing the activation of JNK could increase the mRNA expressions of collagen I, collagen III and MMP9 to cause cardiac fibrosis [43]. Decreasing ROS level under hypoxia condition could suppress JNK activation and the gene expression of CTGF, collagen I and III followed improvement of fibrosis [44]. However, these studies were not focusing on atrial cardiomyocytes. This is the first study that showed ROS/JNK signaling pathway plays an important role in atrial fibrosis. However, the role of the ROS/JNK pathway in different types and stages of atrial remodeling remains to be determined. Furthermore, the profibrotic effect of crosstalk between the ROS/JNK pathway and other hypoxia-dependent signal pathways (e.g., PI3K/Akt) has not yet been explored in our study. It is critical to determine whether the upregulation of HIF-1α is required to induce those profibrotic proteins by silencing HIF-1α prior to hypoxia. However, our results are of value in the study of atrial remodeling in humans and may contribute to further AF studies.

## 4. Materials and Methods

This experimental protocol was approved by the Institutional Animal Care and Use Committee of the Chung Shan Medical University and in compliance with the Guide for the Care and Use of Laboratory Animals (National Institutes of Health publication no. 3040-2, revised 1999).

### 4.1. Hypoxia Stimulation in HL-1 Atrial Myocytes and Drug Preparation

The details of culture of HL-1 atrial cells were described previously [12]. For all the assays, HL-1 cells were incubated in reduced serum medium (2% FBS). For hypoxia, HL-1 cells were placed in a hypoxia chamber (NexBiOxy Inc., Hsinchu, Taiwan) and maintained at 37 °C in a humidified hypoxic atmosphere of 1% O_2_, 5% CO_2_ and 93% N_2_. Controls were maintained in 5% CO_2_ and 95% air at 37°C. The prolyl hydroxylase inhibitor cobalt chloride (CoCl_2_) was prepared from a 100 mM stock solutions. For the investigation of signal pathways, cells were pre-treated with inhibitors for 30 min before exposure to the hypoxic stimulus. SB203580 (a highly specific p38 kinase inhibitor), U0126 (a specific and potent MAPKK/MEK inhibitor) and SP600125 (a potent, selective and reversible c-Jun *N*-terminal kinase [JNK] inhibitor) were purchased from Calbiochem (La Jolla, CA, USA) and 10 mM stock solutions of these inhibitors were made up in DMSO. A stock solution of 100 mM N-acetylcysteine (NAC), a general ROS scavenger, was prepared in buffer (20 mM HEPES, pH 7.0) and pre-treatment of cells with 2 mM NAC for 2 h was used to prevent the formation of ROS.

### 4.2. Transient Transfection

For transient transfections, HL-1 cells were transfected with Lipofectamine 2000 (Invitrogene, Carlsbad, CA, USA) following the manufacturer’s instructions. Typically, 2 × 10^6^ cells were plated in 6-cm cell culture dishes. The next day, cells were transfected with 5 µg/dish of dominant-negative JNK construct and negative control pCDNA3.1. After 48 h, the cells were subjected to hypoxia for the following assay. The dominant-negative JNK construct was a kind gift from Tang Chih-Hsin.

### 4.3. Protein Extracts and Western Blot Analysis

Using Western blotting, we investigated the effects of hypoxia stimulation on the expression of HIF-1α, phosphorylation of different MAP kinases and expression of fibrosis-related proteins (TGF-β1, α-SMA, COL1A and COL3A) in cultured HL-1 atrial cells.

Total cellular protein was extracted from cells cultured under normoxic and hypoxic conditions using a Pro-prep protein extraction solution (iNtRON Biotechnology, Gyeonggi-do, Korea). The homogenates were centrifuged at 15,000× *g* for 20 min at 4 °C and stored at −20 °C. Nuclear fractions were obtained by disrupting the frozen homogenates using a Kontes Dounce Homogenizer with hypotonic lysis buffer (10 mM HEPES, 1.5 mM MgCl_2_, 10 mM KCl, 0.5 mM DTT, 0.05% Igepal, pH 7.9). The lysate was centrifuged at 900× *g* for 5 min then the nuclear pellet was subjected to lysis by adding Pro-prep protein extraction solution. Protein concentrations were determined using the DC Protein Assay (BioRad Laboratories, Inc., Hercules, CA, USA) and 25 μg of soluble protein per sample was analyzed by electrophoresis in 8% or 12% polyacrylamide gels containing SDS (SDS-PAGE). After electrophoresis, the samples were transferred to polyvinylidene difluoride (PVDF) membranes (Millipore, Bedford, MA, USA).

The membranes were blocked with 3% bovine serum albumin (BSA) for 1 h at RT and incubated with antibody against either HIF-1α (Santa Cruz Biotechnology, Santa Cruz, CA, USA), α-SMA (Santa Cruz Biotechnology, Santa Cruz, CA, USA), TGF-β1 (Geneway Biotech, San Diego, CA, USA), COL1A (Abcam, Cambridge, MA, USA), p-c-Jun, p-ATF2, p-SEK/MKK4 (Cell Signaling Technology, Danvers, MA, USA), p38α, p38 MAPK (pT180/pY182), pan-JNK/SAPK, JNK/SAPK (pT183/pY185), ERK, or ERK (pT201/pY204) at 4 °C overnight. A duplicate membrane was probed with anti-β-actin (Geneway Biotech, San Diego, CA, USA). The membranes were then incubated with a species-specific horseradish peroxidase labeled secondary antibody (BD Biosciences: 1:5000) (BD Biosciences, San Jose, CA, USA) (1:5000) for 1 h at 37 °C, thoroughly washed with TBST between each step and stained with ECL reagents to visualize peroxidase activity on ImageQuant LAS4000 mini digital imaging system (Fujifilm-GE Healthcare, Tokyo, Japan). For quantification purposes, blots were scanned using a luminescence imager (LAS-1000 Image Analyzer, Fujifilm, Berlin, Germany) and analyzed with FluorChem image software (Alpha Innotech, San Leandro, CA, USA).

### 4.4. Measurement of Reactive Oxygen Species (ROS)

Intracellular ROS production was measured by using a cell-permeable fluorescent probe, 2′,7′-dichlorofluorescein diacetate (DCFH-DA) (Merck, Darmstadt, Germany). Cells were seeded in 96-well plates and exposed to DCFH-DA for 30 min, then treated with NAC for 2 h followed by hypoxia. After incubation with 20 µM DCFH-DA for 30 min, the fluorescence was read at 485/530 nm (excitation/emission) on a Synergy™ HT Multi-Mode Microplate Reader and the readings were normalized by cell viability measured using a Cell Counting Kit-8 (CCK-8; Sigma) (Sigma-Aldrich, Saint Louis, MO, USA).

### 4.5. Immunohistochemical Analysis

For immunofluorescence, 10^4^ HL-1 cells cultured on gelatin-fibronectin coated 8-well chamber slides were serum starved for 24 h, stimulated by incubation in a hypoxic atmosphere of 1% O_2_, fixed by 4% paraformaldehyde in phosphate-buffered saline (PBS) for 15 min, permeabilized in 0.5% Triton X-100 for 10 min, blocked with 1% BSA in PBS for 1 h at room temperature, stained 2 h at room temperature with antibody against COL1A (Abcam, Cambridge, UK) and COL3A (Santa Cruz Biotechnology, Santa Cruz, CA, USA) both diluted 1:100 in 1% BSA in PBS, washed with PBS twice and incubated for an additional 60 min at room temperature with goat anti-mouse IgG or goat anti-rabbit IgG conjugated with fluorescein isothiocyanate (FITC) at 1:200 dilution in 1% BSA in PBS. Images were acquired with a DP72 digital camera on a DP2-BSW microscope and analyzed by digital camera software. As a negative control, HL-1 cells were incubated with the secondary antibody alone. To measure staining intensity, digital images were sectioned into their basal and apical compartments and densitometric analysis of immunofluorescent staining intensity was performed using an image routine involving the number of pixels and label (area of label) in Image J software (National Institutes of Health, Bethesda, MD, USA). Uniform microscope and laser settings were used for each experimental condition. The number of pixels with label (area of label) changes is expressed as a percent increase or decrease and the magnitude of these changes was statistically evaluated.

### 4.6. Statistical Analysis

SPSS 13.0 was used in the statistical analysis. Statistical significance of differences between treatments throughout this study were analyzed by One-way ANOVA and followed up by Dunnett’s multiple comparison post hoc test. The *p*-values indicating statistical significance are given in the respective figure caption. A *p* value < 0.05 was considered statistically significant.

## 5. Conclusions

Together our data demonstrate that hypoxia-induced profibrotic proteins expression in cultured HL-1 atrial cells is mediated by the activation of ROS and the JNK pathway. The results of the present study suggest that the novel ROS/JNK/ATF2 signaling pathway mediates the fibrogenic effect of hypoxia stress and potentially represents an important target of AF therapy.

## Figures and Tables

**Figure 1 ijms-22-03249-f001:**
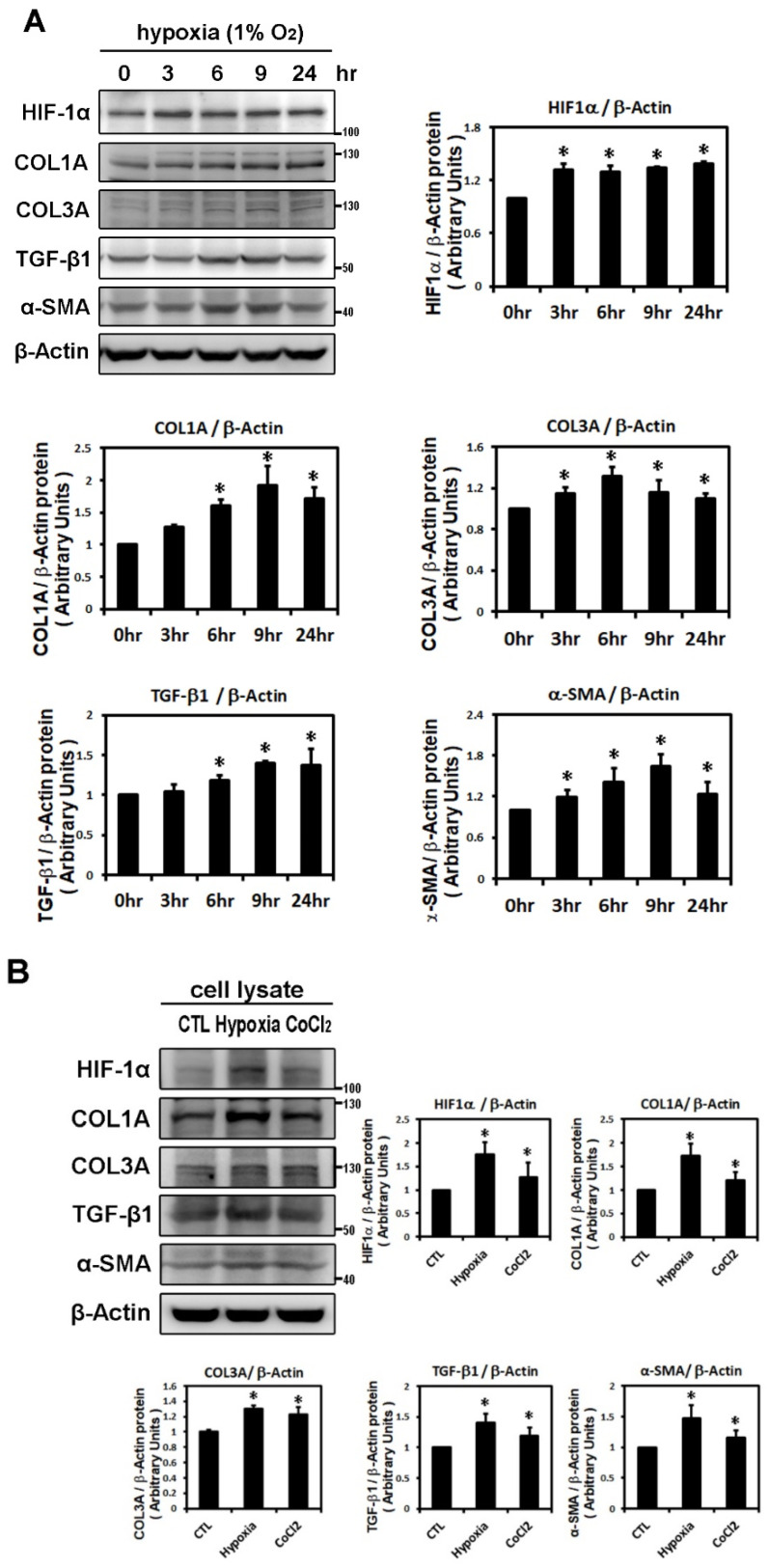
Effects of hypoxic treatment on the expression of HIF-1α and profibrotic proteins. Expression of the indicated proteins in cellular homogenates was measured by Western blot. Each lane contained a total of 25 μg of protein and the blots were probed with the indicated antibody. All data were normalized to the β-actin loading and blotting control. Bar graphs indicate band intensity as determined by densitometry and the fold increase in intensity over control. Panel **A**: Exposure of cultured HL-1 cells to hypoxia (1% O_2_) for the indicated times resulted in time-dependently increased protein levels of HIF-1α (maximum reached at 3 h), COL3A (maximum reached at 6 h) and TGF-β1, α-SMA, COL1A (maxima reached at 9 h). Panel **B**: Representative Western blots and quantitative analysis of protein levels of HIF-1α and fibrosis-related proteins (COL1A, COL3A, TGF-β1, α-SMA) in HL-1 cells treated with hypoxia (1% O_2_) or CoCl_2_ (10^−4^ M) for 6 h. The data are mean ± SEM (*n* = 4 per group, * *p* < 0.05 vs. control) from four independent experiments.

**Figure 2 ijms-22-03249-f002:**
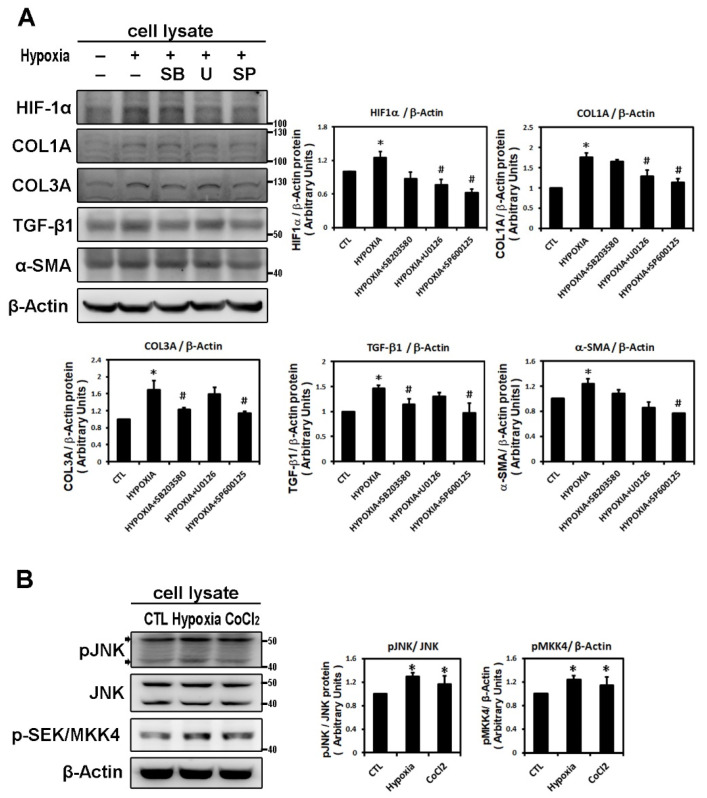
Panel **A**: Effect of MAPK inhibitors on the expression of hypoxia-induced HIF-1α and profibrotic proteins. Whole cell lysates of HL-1 cells pre-incubated with SB203580 (10^−5^ M), U0126 (10^−5^ M), SP600125 (10^−5^ M) or DMSO as a vehicle for 30 min, then treated with hypoxia stress (1% O_2_) for 6 h, were analyzed by Western blots to determine the levels of HIF-1α and fibrosis-related proteins (COL1A, COL3A, TGF-β1, α-SMA) expression. The bar graph shows the value of each sample relative to that of control cells that received vehicle only. The data are mean ± SEM (*n* = 4 per group, * *p* < 0.05 vs. control; # *p* < 0.05 vs. hypoxia) from four independent experiments. Panel **B**: Effect of hypoxic treatment on phosphorylated JNK signals. Representative Western blots and protein levels of JNK, phosphorylated JNK and SEK/MKK4 in HL-1 cells treated with hypoxia (1% O_2_) or CoCl_2_ (10^−4^ M) for 6 h. Panel **C**: Exposure of cultured HL-1 cells to hypoxia (1% O_2_) for the indicated times resulted in time-dependently increased protein levels of phosphorylated JNK and SEK/MKK4. The data are mean ± SEM (*n* = 4 per group, * *p* < 0.05 vs. control) from four independent experiments.

**Figure 3 ijms-22-03249-f003:**
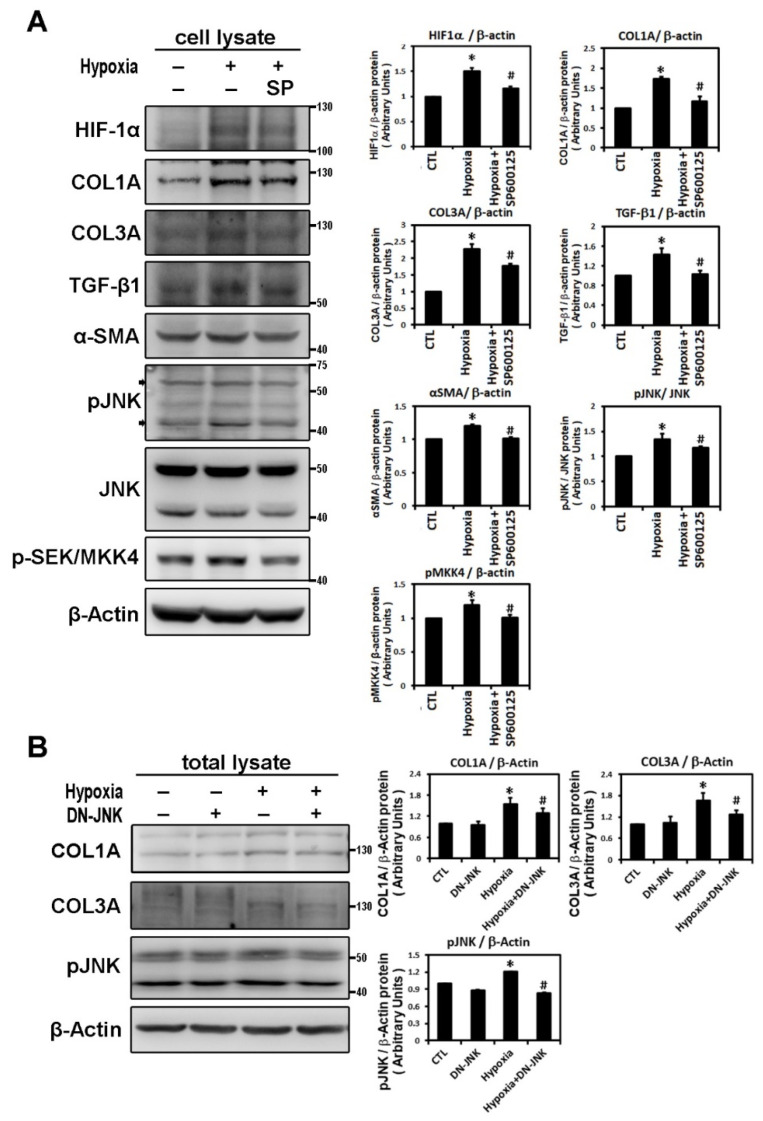
Hypoxia activates the JNK pathway to modulate fibrosis-related proteins expression. Panel **A**: Effect of JNK pathway inhibitor on the hypoxia-induced expression of indicated protein. Representative Western blots and levels of HIF-1α, fibrosis-related proteins (COL1A, COL3A, TGF-β1, α-SMA) and phosphorylated JNK and SEK/MKK4 in HL-1 cells that were pre-incubated with SP600125 (10^−5^ M) or DMSO as a vehicle for 30 min, then treated with hypoxia stress (1% O_2_) for 6 h. The bar graph shows the value of each sample relative to that of control cells that received vehicle only. The data are mean ± SEM (*n* = 4 per group, * *p* < 0.05 vs. control; # *p* < 0.05 vs. hypoxia) from four independent experiments. Panel **B**: Dominant-negative-JNK (DN-JNK) gene transfection attenuated hypoxia-induced COL1A, COL3A and *p*-JNK expression in HL-1 cells. Representative Western blots and COL1A and COL3A levels in HL-1 cells transfected with vehicle pCDN3 (CTL) or DN-JNK for 24 h before 6h-hypoxia treatment. The data are mean ± SEM (*n* = 3 per group, * *p* < 0.05 vs. control; # *p* < 0.05 vs. hypoxia) from three independent experiments.

**Figure 4 ijms-22-03249-f004:**
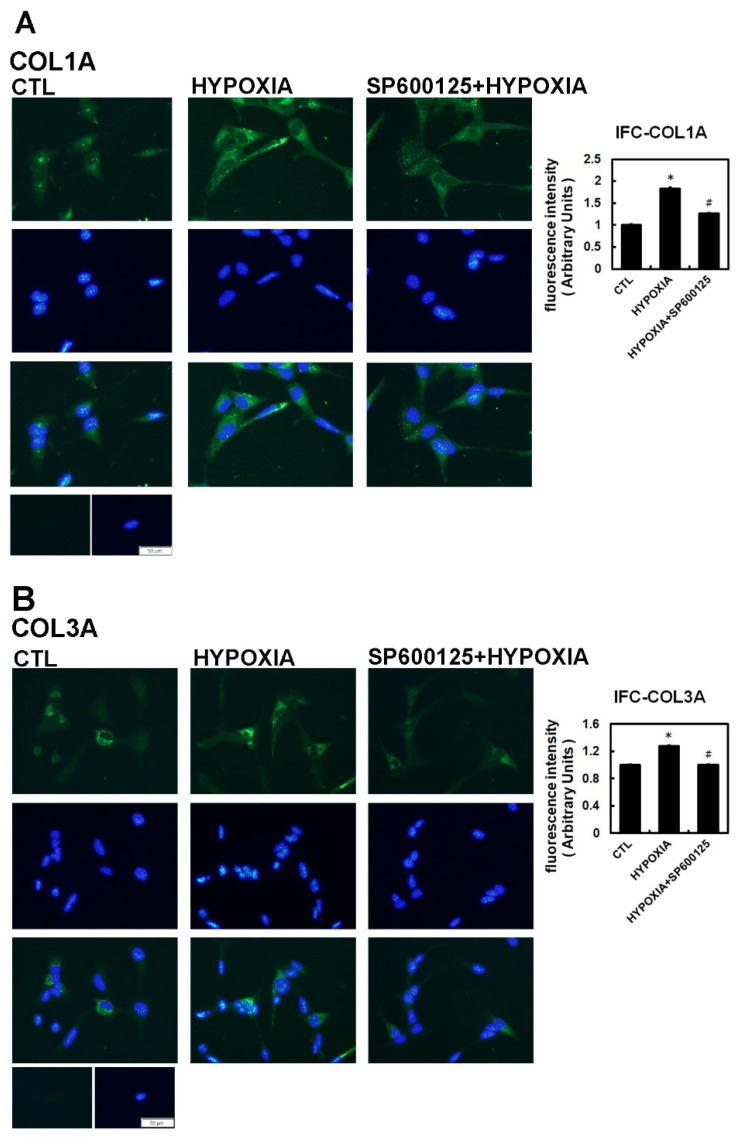
Immunohistochemistry assay of hypoxia-treated HL-1 cells. The HL-1 cells were pre-treated with inhibitors SP600125 (10^−5^ M) or DMSO as a vehicle for 30 min, treated with hypoxia stress (1% O_2_) for 6 h, fixed and subjected to immunostaining with anti-COL1A (Panel **A**) and anti-COL3A antibody (Panel **B**). Quantification of the number of pixels with intracellular label was determined digitally from the fluorescence microscopic images. All indicated fibrotic marker protein densities were significantly reduced in all the cells pre-treated with SP600125 compared to those treated by hypoxia only. Values are the arithmetic means of at least three independent experiments and change to the numbers of pixels are expressed as percent increase or decrease compared to the control level (DMSO only). The percentage of fluorescing cells was calculated per condition as described in Methods. The scale bar corresponds to 50 µM. The data are mean ± SEM (*n* > 100, * *p* < 0.01 vs. control; # *p* < 0.05 vs. hypoxia).

**Figure 5 ijms-22-03249-f005:**
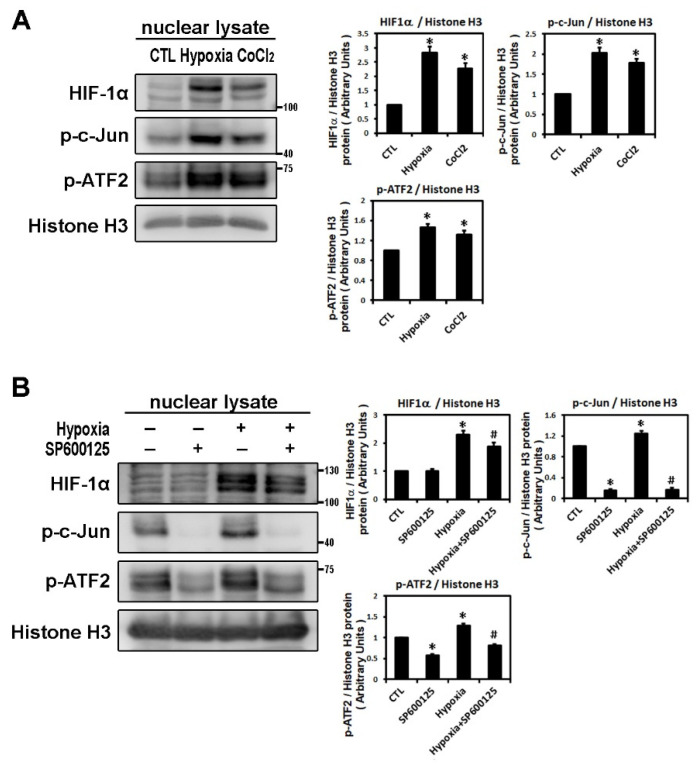
Analysis of nuclear fraction from hypoxia-treated HL-1 cardiomyocytes that modulated by JNK transduction pathway. Panel **A**: Nuclear lysates were analyzed by Western blots to determine levels of HIF-1α, *p*-c-Jun, p-ATF2 in HL-1 cells under the control condition (no treatment), treatment with CoCl_2_ (10^−4^ M) for 6 h, or hypoxia (1% O_2_) for 6 h. The data are mean ± SEM (*n* = 4 per group, * *p* < 0.05 vs. control) from four independent experiments. Panel **B**: Representative Western blots and levels of HIF-1α, p-c-Jun, p-ATF2 in the nuclear lysate of HL-1 cells pre-treated with SP600125 (10^−5^ M) or DMSO as a vehicle for 30 min, then treated with hypoxia stress (1% O_2_) for 6 h. The bar graph shows the value of each sample relative to that of control cells under normoxia (vehicle only) or hypoxic condition. The data are mean ± SEM (*n* = 3 per group, * *p* < 0.05 vs. control; # *p* < 0.05 vs. hypoxia) from three independent experiments.

**Figure 6 ijms-22-03249-f006:**
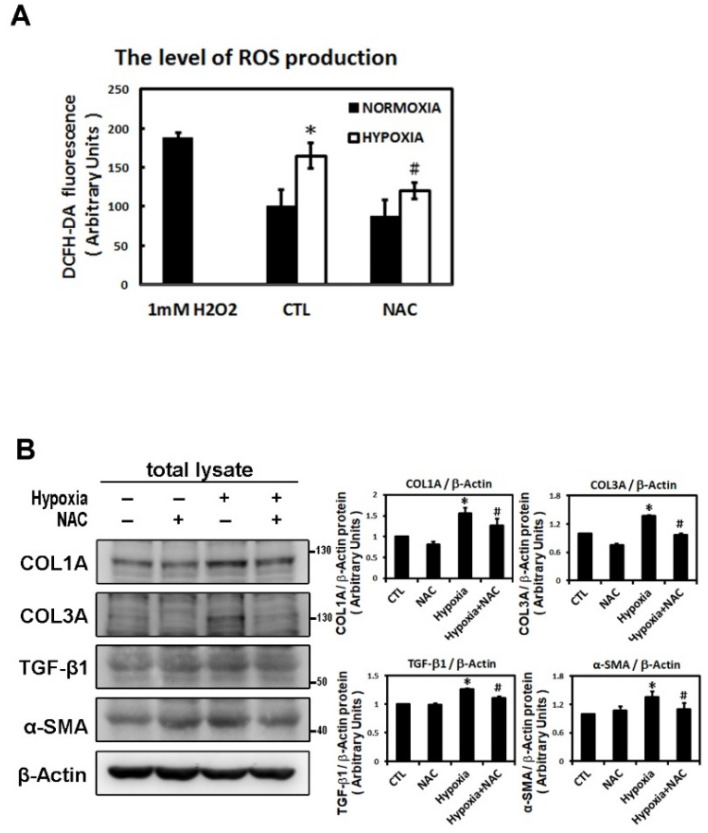
Reactive oxygen species (ROS) module hypoxia-induced HIF-1α, JNK signal pathway and fibrosis-related proteins expression in HL-1 cells. Panel **A**: The ROS levels were assessed with dichlorofluorescin diacetate (DCFH-DA) in HL-1 cells that were pre-incubated with NAC (2 mM) or vehicle only (CTL) for 2 h, then subjected to normoxia (5% CO_2_ and 95% air) or hypoxia (1% O_2_) stimulation for 6 h. The bar graph shows the value of each sample relative to that of control cells under normoxic (vehicle only) or hypoxic conditions. The data are mean ± SEM (*n* = 4 per group, * *p* < 0.05 vs. control; # *p* < 0.05 vs. hypoxia) from four independent experiments. Panel **B** and **C**: Representative Western blots and levels of fibrosis-related proteins (COL1A, COL3A, TGF-β1, α-SMA), JNK, phosphorylated JNK and MKK4 for whole cell lysate in HL-1 cells pre-treated with or without NAC (2 mM) for 2 h, then treated under hypoxic (1% O_2_) or normoxic conditions for 6 h. The data are mean ± SEM (* *p* < 0.05 vs. control; # *p* < 0.05 vs. hypoxia) from at least three independent experiments. Panel **D**: Representative Western blots and levels of HIF-1α, fibrosis-related proteins (COL1A, COL3A, TGF-β1, α-SMA) in nuclear lysates of HL-1 cells pre-treated with or without NAC (2 mM) for 2 h, then treated under hypoxic stress (1% O_2_) or normoxic conditions for 6 h. The bar graph shows the value of each sample relative to that of control cells under normoxic (vehicle only) or hypoxic conditions. The data are mean ± SEM (* *p* < 0.05 vs. control; # *p* < 0.05 vs. hypoxia) from at least three independent experiments.

**Figure 7 ijms-22-03249-f007:**
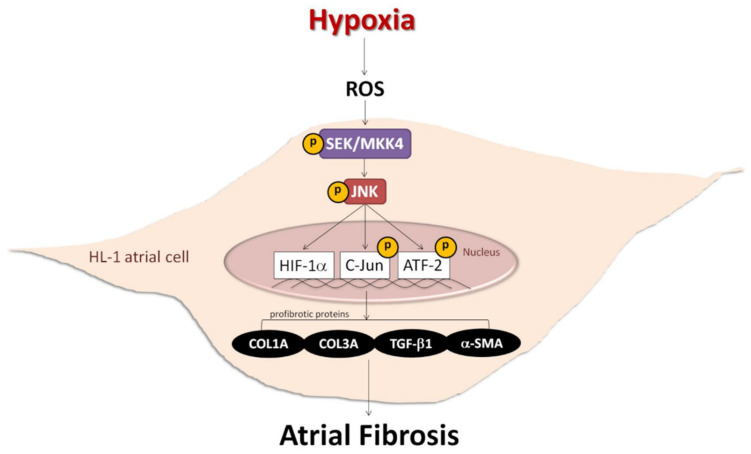
Schematic diagram of hypoxia promoting atrial fibrosis via the ROS/JNK pathway and downstream activation of nuclear c-Jun/ATF2 phosphorylation to up-regulate the expression of fibrosis-related proteins in HL-1 cells.

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
