# Peer review of "Role of the ROS-JNK Signaling Pathway in Hypoxia-Induced Atrial Fibrotic Responses in HL-1 Cardiomyocytes"

_ijms, 2021, doi:10.3390/ijms22063249_

Round 1

Reviewer 1 Report

The paper titled "Role of the ROS-JNK Signaling Pathway in Hypoxia-Induced Atrial Fibrotic Responses in HL-1 Cardiomyocytes" by Tsai et al. uses HL-1 cardiomyocytes to reveal that hypoxia induced fibrotic marker proteins via ROS-JNK signaling pathway. The authors proposed that ROS-JNK signaling pathway could be a potentially therapeutic target for atrial fibrillation because fibrosis is an important factor of development of atrial fibrillation. Although the findings of the present study are interesting and important, the several data shown in this study were unfortunately not convincing (please see the following comments). Therefore, there are a number of concerns, which should be addressed before any consideration for publication.

Major Comments

  1. In abstract, the authors write, "Our results suggest that a novel ROS/JNK signaling pathway may provide a potential therapeutic target for the treatment of atrial fibrillation." This study shows that the ROS/JNK signaling pathway is a key factor in development of fibrosis in the atrium. However, the results in the present study did not show that fibrosis itself could be prevented. In my opinion, the phrase "therapeutic target for the treatment" would be a bit overblown.
  2. In Figure 1A, the authors describe that the expression levels of Col1A, COl3A, TGFβ1, and αSMA proteins were declined 24 hours after hypoxic treatment, whereas the expression levels of HIF1a protein remained high even after 24 hours. Is there any explanation why the expression of profibrotic proteins is down-regulated earlier than HIF1a protein?
  3. Figure 1B shows a 25-fold increase in COL3A protein expression with hypoxia (1% O2), while Figure 1A shows only a 1.5-fold increase in COL3A with the same 6-hour hypoxic condition. I am afraid that there a significant difference between these results (Figure 1A and 1B). Accordingly, the expression pattern of COL3A in the Western blotting membranes is not consistent in the entire study. I found a single clear band of COL3A in Figure 1B and Figure 2A. However, some Figures showed many non-specific bands or unclear band of COL3A. I also realized the same problem of HIF1a For example, in Figure 5, the signals in the HIF1a are quite different between Figure 5A and Figure 5B. I am afraid that these undermine the reliability of the entire study.
  4. I think that ROS-JNK signaling pathway should be activated earlier than 6 hours in response to hypoxia. I strongly recommend that the authors should provide the data of the time-course in ROS-JNK signaling pathway in HL-1 cardiomyocytes in response to hypoxia.
  5. As the authors pointed out in the study limitation, it is still unclear that hypoxia indeed would induce ROS/JNK signaling pathway to promote fibrosis in the atrium in in vivo study, which would strengthen the present study.

Minor Comments

  1. In Figure 3A, the Hif1a/β-Actin protein expression ratio should be calculated to be 1.0 for control.
  2. The x-axis label in Figure 3A should be Hypoxia+SP600125 instead of SP600125 only.
  3. The x-axis label in Figure 4 should be better consistent with the other figures, Hypoxia+SP600125 instead of SP+Hypoxia.
  4. It would be strange that the expression levels of p-c-Jun and p-ATF2 proteins in Figure 5B are not significantly different between CTL and SP600125 groups.
  5. it seems that the expression levels of pMKK4 protein in Figure 6C is clearly increased in NAC treatment.

Reviewer 2 Report

This article by Tsai et al. investigated the roles of ROS and JNK signaling in hypoxia-induced profibrotic response in the atrial cardiomyocyte cell line HL1 (SC). Atrial fibrosis is an important contributor to the development of the atrial substrate. Therefore, it is critical to understand the mechanisms of how hypoxia leads to fibrosis in atrial cardiomyocytes. The authors demonstrated that hypoxia treatment of HL1 cells leads to generation of ROS and activation of the JNK signaling pathway, which results in phosphorylation of c-Jun and ATF2, nuclear translocation of HIF1a, and increased protein expression of fibrotic proteins. Overall, the manuscript has properly designed experiments, accurately interpreted data, and clinically relevant findings. Nevertheless, there are some concerns and suggestions:

  1. It will be very nice if the authors can include a model figure at the end to summarize findings in this manuscript and illustrate how hypoxia may lead to fibrosis in the atria.
  2. I understand that the authors do not intend to investigate in the current study how c-Jun, ATF2, and HIF1a activate fibrotic gene expression, and they actually pointed that out in the Discussion. But that is a critical part of the mechanisms studied here and it will be very interesting and highly important for the authors to at least discuss possible mechanisms based on the literature. For example, does c-Jun, ATF2, and/or HIF1a activate transcription of collagen/TGFbeta/aSMA expression? Even if nothing is available in the literature, the authors can at least check whether promoter regions of these fibrotic genes contain c-Jun, ATF2, and/or HIF1a binding sites.
  3. Some control experiments/additional labeling are needed:
    1. Figure 3B: there need to be positive controls showing whether the DN-JNK is actually expressed (RNA or protein) and whether JNK signaling is inhibited.
    2. Figure 4: isotype or no IgG control missing for COL1A and COL3A IF.
    3. All western blots: need to indicate where size markers are.
    4. Figure 2B: why two bands of JNK? No explanation in results or legend.
  4. There are some language issues in the current version of the manuscript. The results section is unnecessarily lengthy and it will be great if the authors and write the manuscript in a succinct way. It is strongly recommended that academic writing professionals be consulted.

Round 2

Reviewer 2 Report

Accept as is.